# Darkness promotes dishonesty in a coin toss task: A pre-registered conceptual replication of Experiment 1 of Zhong, Bohns, and Gino (2010)

**Huanxu Liu** [1]*, **Yuning Zhou**[2], **Yuki Yamada**[3]

**1** Graduate School of Human-Environment Studies, Kyushu University, Fukuoka, Fukuoka, Japan,
**2** Department of Psychiatry, The First Hospital of China Medical University, China Medical University,
Shenyang, Liaoning, China, **3** Faculty of Art and Science, Kyushu University, Fukuoka, Fukuoka, Japan

* ryuukansyo@gmail.com

Darkness promotes dishonesty in a coin toss task:
A pre-registered conceptual replication of
Experiment 1 of Zhong, Bohns, and Gino (2010).
PLoS ONE 18(12): e0294484. https://doi.org/

ISRAEL

**Data Availability Statement:** The datasets
analyzed in this study are available from the OSF
repository (osf.io/bvqsn).

## Abstract

An earlier study suggested that individuals tend to be more dishonest in darker environments, this phenomenon was attributed to an increase in participants' perceived anonymity. However, instead of using quantifiable measurements, the original experiment depended on the experimenter's subjective observation to construct different brightness conditions, which led to a less precise understanding of the phenomenon. Additionally, the task used in the original experiment has recently been criticized as being unsuitable for dishonesty-detection studies. This study addressed these concerns to retest the effect of brightness on dishonest behavior. This study employed lux as a unit to accurately control the brightness within the experiment room. Moreover, the coin toss task which is frequently employed in dishonesty-detection experiments, was utilized instead of the task in the original experiment. The findings revealed that despite altering the content of the task, dishonesty increased in the dark condition. Contrary to the findings in the original experiment, however, the results did not substantiate that perceived anonymity was the driving factor of the effect of brightness on dishonesty. This discrepancy suggests that further empirical considerations are warranted to unravel the underlying mechanisms.

## Introduction

The potential adverse consequences of dishonesty in daily life have catalyzed extensive research in psychology. For instance, previous studies in social psychology have explored the influence of dishonesty on moral judgments and behavioral trust [1], the effect of the presence of observers on dishonesty [2], and dishonesty in groups [3]. Another context in personality psychology, it has also investigated many issues, such as the moderation effect of personality on the relationship between the ability to assess fraud risk and detect fraud [4], and the role of personality in fraud victimization [5]. Physiological psychology has also contributed to the understanding of this issue, with the studies about the relation between physiological indicators (such as hormones and facial thermal variations) and dishonesty behavior [6–8]. Even

**Funding:** This study was financially supported by JST SPRING (https://www.jst.go.jp/jisedai/en/index.html), Grant Number JPMJSP2136 (HL), Grants-in-Aid for Scientific Research of the Japan Society for the Promotion of Science (https://www.jsps.go.jp/j-grantsinaid/index.html), Grant Numbers 22K18263, 21H03784, and 20H04581 (YY). The funders had no role in the study design, data collection and analysis, publication decision, or manuscript preparation.

**Competing interests:** The authors have declared that no competing interests exist.

though these studies occasionally report null results, they underlined the importance of dishonesty as a valuable research topic.

While the preponderance of psychological research on dishonesty has focused on social and individual factors, Zhong et al. (2010) introduced a novel perspective, highlighting the role of environmental factors, such as room darkness, in influencing dishonest behavior [9]. Zhong et al. demonstrated that the darkness engendered an increase in dishonest behavior and attributed this influence to alterations in perceived anonymity. Perceived anonymity is closely tied to the environment's actual anonymity but also may be swayed by cognitive bias. Zhong et al. (2010) emphasized that while the actual anonymity in their experiment does not vary with the darkness of the room, participants' cognitive biases (caused by the phenomenological experience of impaired vision in the dark) led them to believe they were less observable in the dark [9]. This amplified their perceived anonymity, thereby escalating dishonesty—a phenomenon they termed "illusory anonymity." This explanation is supported by the "Three Principles to REVISE People's Unethical Behavior" proposed by Ayal et al. (2015) [10], which point out that increased visibility potentially reduces anonymity, enabling better surveillance and thereby deterring dishonesty. Furthermore, anonymity has been linked to concepts like decision observability, with studies reporting that increased decision observability (i.e., reduced anonymity) may stimulate pro-environmental and pro-social behaviors (e.g., [11]). Consequently, it seems reasonable that Zhong et al. (2010) explained the observed increase in dishonest behavior in dark conditions was due to a heightened sense of anonymity [9].

However, several reasons led us to conduct a replication of Experiment 1 of Zhong et al. (2010). First, Zhong et al. (2010) considered that the actual anonymity was not altered. They said that "although the room in the experimental condition was darker than the one in the control condition, participants had no trouble seeing and identifying each other" [9, p. 314]. By contrast, we considered that the visibility was altered at least to some degree by their method, for similar sized rooms of 15 ft x 14 ft, there were 12 fluorescent lights in light conditions and 4 fluorescent lights in dark conditions. Although the dishonesty in the task (the Matrix Task from [12]) of their experiment could not be detected simply by observation, the experimenters could detect dishonesty attempts by observing the participants' expressions. In this instance, the observation by other parties, such as the experimenter, could compromise the participants' temporary anonymity (i.e., the state when the participants performed the task without the observation), indicating that the actual anonymity within the experimental context is potentially subject to change. Another reason is that participants' perceived anonymity was important in their arguments, however, we did not find any relevant results reported in the experiment of dishonesty (Experiment 1). Zhong et al. (2010) did not examine any measures related to perceived anonymity, thus it is unclear whether the mechanism in which room darkness leads to cheating really involves perceived anonymity [9]. That is, the effect of anonymity has insufficient plausible grounds to support the conclusions of the original study. Further empirical evidence is required to substantiate this claim. The third reason is that the Matrix Task, as discussed, is ill-suited for studying dishonesty [13]. Hence, despite the widespread use of the Matrix Task, we aim to validate the effect of darkness on dishonesty by employing an alternative task. For these reasons, we considered it necessary to conduct further experiments to explore the relationships between anonymity, darkness, and dishonesty. Compared to the original experiment, we adopted the individual experiment approach, i.e., created an environment where the actual anonymity is consistently zero. We then elicited participants' illusory anonymity by dimming the room's brightness, then investigated the perceived anonymity and used another widely used task to assess the dishonest behaviors. In summation, our refinements provide several key advantages: first, they can control the variability of actual anonymity. Second, they can provide evidence for the first time on perceived anonymity in the

observed phenomenon. Third, they facilitate a discussion regarding the potential generalizability of the phenomenon, especially under tasks more suitable for detecting dishonest behavior.

We planned an experiment with a between-subjects design which has 1 factor (brightness) with 2 levels (light and dark). Based on the results of the original experiment [9], we anticipated a significantly higher frequency of dishonest behavior in the dark condition compared to the light condition, leading us to formulate the following hypothesis:

Hypothesis (H1): Darkness will promote dishonest behavior.

## Methods

### Ethical statement

The experiment was conducted in accordance with guidelines from the Declaration of Helsinki (2013). The ethics committee of the First Hospital of China Medical University approved the protocol (approval number: 2021-388-2). Before the experiment, each participant provided written informed consent, indicating voluntary participation with an understanding that they retained the right to withdraw from the experiment at any given time.

### Power analysis and participants

The required sample size was calculated in G*power 3.1.9.3. In G*Power, we set the significance level $\alpha$ = 0.05, power level 1-$\beta$ = 0.8, and the effect size $d$ = 0.5 (medium effect; [14]) for the $t$-test. The required total sample size is 102, with 51 participants in each group. Participants were undergraduate and graduate students at China Medical University and staff at the First Hospital of China Medical University. We recruited the participants while excluding those who violated the exclusion criteria (See "Data exclusion"). When we received 102 valid data, we stopped the data collection.

### Procedure

The experiment was conducted in a small room within the First Hospital of China Medical University (Fig 1). Firstly, participants provided written informed consent. Then, participants were randomly assigned to one of the two brightness conditions. To observe the effect of brightness more exactly, we differentiated the brightness in two conditions to the greatest extent feasible within a reasonable range. The light condition was calibrated to over 1000 lux, while the dark condition was set to less than 10 lux, as measured by a lux meter (Deli DL333205). Brightness was manipulated using a lamp with adjustable brightness (MKD MT-L200).

The participants were initially informed that they were assisting with research on paranormal phenomena, and they were asked to provide informed consent. After introducing the experiment, the experimenter sat with his/her back to the participants in another corner of the room, separated by a row of lockers. Then, the participants started the "coin toss task". This task is an adjusted version of the paradigm presented by Fischbacher and Föllmi-Heusi (2013) for dishonest detection [15]. The paradigm was widely used and Schild et al. (2021) asserted its external validity recently [16]. In the present study, several coins were provided on a table (Fig 1), and participants were free to select any one coin. Then, participants were asked to flip the coin 10 times and report the number of "tails" outcomes as their score. For every "tail", participants could receive 3 Yuan (CNY) as a reward. Participants were instructed these rules in advance. Given that the experimenters were unaware of the participants' true scores, this setup

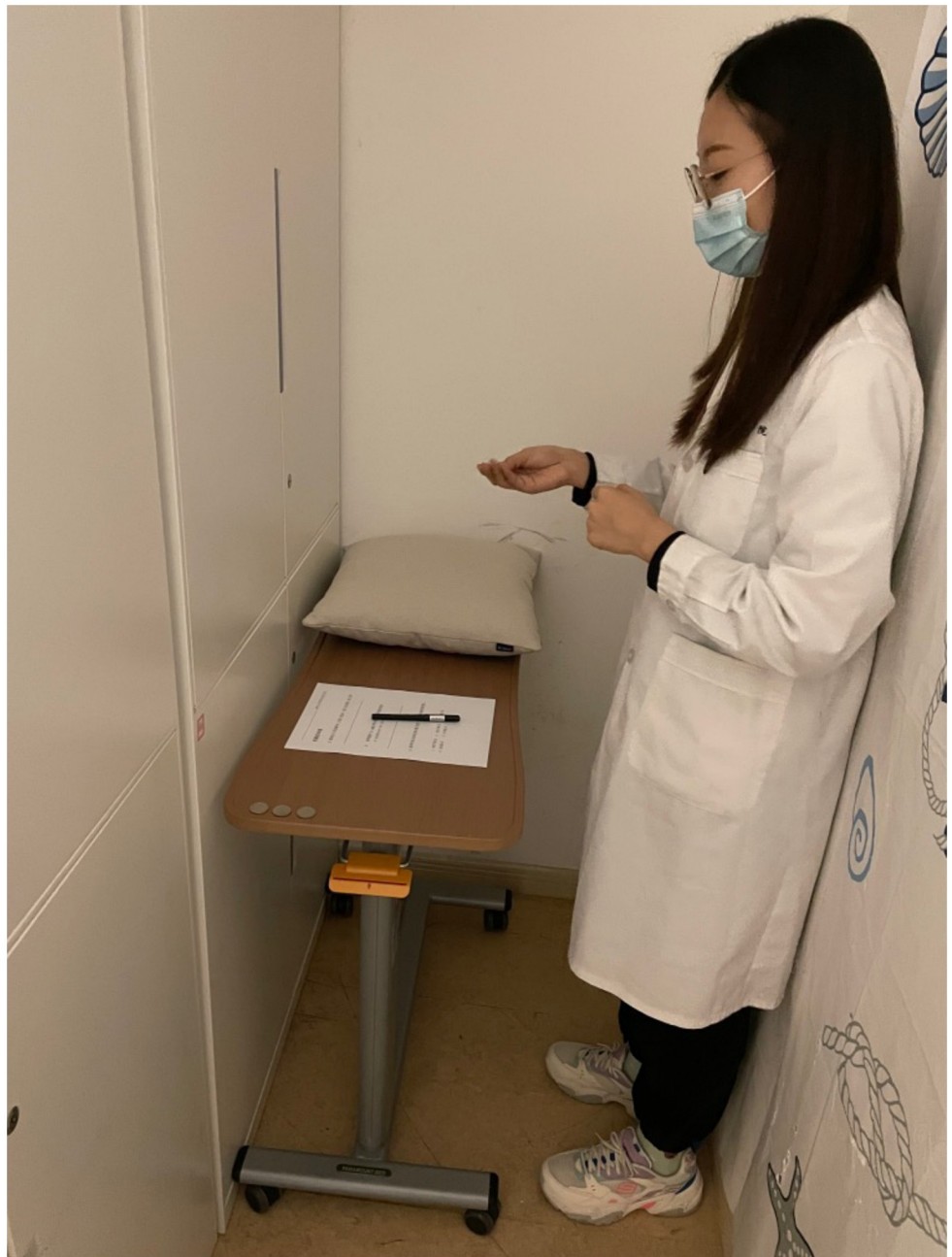

**Fig 1. The coin toss task.**

provided an opportunity for the participants to be dishonest to receive more rewards. We utilized the scores reported by participants as a measure of the degree of dishonesty. See "Data Analyses" for more details.

After participants reported their scores, they were asked to evaluate the room brightness by using a magnitude estimation method (from 0: *not light at all* to 100: *very strongly light*) and the perceived anonymity by using a 5-point Likert scale (1: *not anonymous* to 5: *very anonymous*).

Participant recruitment commenced on October 12, 2021, and the experiment was carried out through December 14, 2021.

## Data exclusion

Participants were excluded if they unable to complete the task properly, did not provide adequate data, determined the real purpose during the experiment, or demanded the removal of their data.

## Data analyses

A *t*-test was used to test whether there was a significant difference between the mean number of "tails" in the two brightness conditions. Besides, the *t*-test was also used to test whether there was a significant difference between the mean number of "tails" reported by the participants in the two conditions and chance level (i.e., 5) to confirm if the dishonesty occurred. If there is a significant difference between the reported scores and the chance level, it has been considered that the participants were dishonest [17, 18]. According to the protocol, we would also use the two one-sided tests (TOST) to test if the effect size was significantly equivalent to 0 when we did not obtain a significant difference between the two conditions in the first analysis mentioned above. We would set the smallest effect size of interest (SESOI) as $d = 0.5$ [14, 19].

## Pre-registration

The protocols above were registered on Open Science Framework (OSF) on October 11, 2021 (https://osf.io/nkjq5) and April 3, 2022 (https://osf.io/p54tj). The following content was written based on data obtained after the pre-registration.

## Results

A total of 105 participants were recruited to collect the pre-registered number of 102 participants while excluding those who violated the exclusion criteria. Three participants were excluded from the analyses because they failed to provide adequate data (due to their movement or touching of the desk lamps, the lighting conditions were altered). Data from the remaining 102 participants (female 68.63%, male 29.41%, no response 1.96%, $M_{age} = 23.73$, $SD = 6.65$) comprised the final dataset for the analyses. Analyses were conducted by using jamovi software (2023, Version 2.3.21.0; https://www.jamovi.org/).

Firstly, for the manipulation check, we compared the perceived brightness in two conditions. The perceived brightness in the light condition was significantly higher than in the dark condition ($t(100) = -8.86$, $p < .001$, Cohen's $d = -1.75$).

Next, we performed a *t*-test to compare the number of "tails" that participants reported and perceived anonymity in two conditions (Fig 2). The mean score reported in dark condition was 5.96 ($SD = 1.48$; $Mdn = 6$); the mean score reported in light condition was 5.39 ($SD = 1.56$; $Mdn = 6$). Participants in the dark condition reported a significantly higher score than the light condition ($t(100) = 1.885$, $p = .031$, Cohen's $d = 0.373$). On the other hands, the mean perceived anonymity in dark condition was 4.37 ($SD = 0.87$), while that in light condition was 4.24 ($SD = 1.11$). The difference between the perceived anonymity in the two conditions was not significant ($t(100) = 0.696$, $p = .244$, Cohen's $d = 0.138$). For this reason, we appended a TOST to test if the effect size could be regarded as 0. The results of the TOST were significant (TOST Upper: $t(100) = 3.22$, $p < .001$, TOST Lower $t(100) = -1.83$, $p = .035$, equivalence bounds (raw): Low = -0.498, High = 0.498), which means that the effect of darkness on perceived anonymity was significantly equivalent to 0.

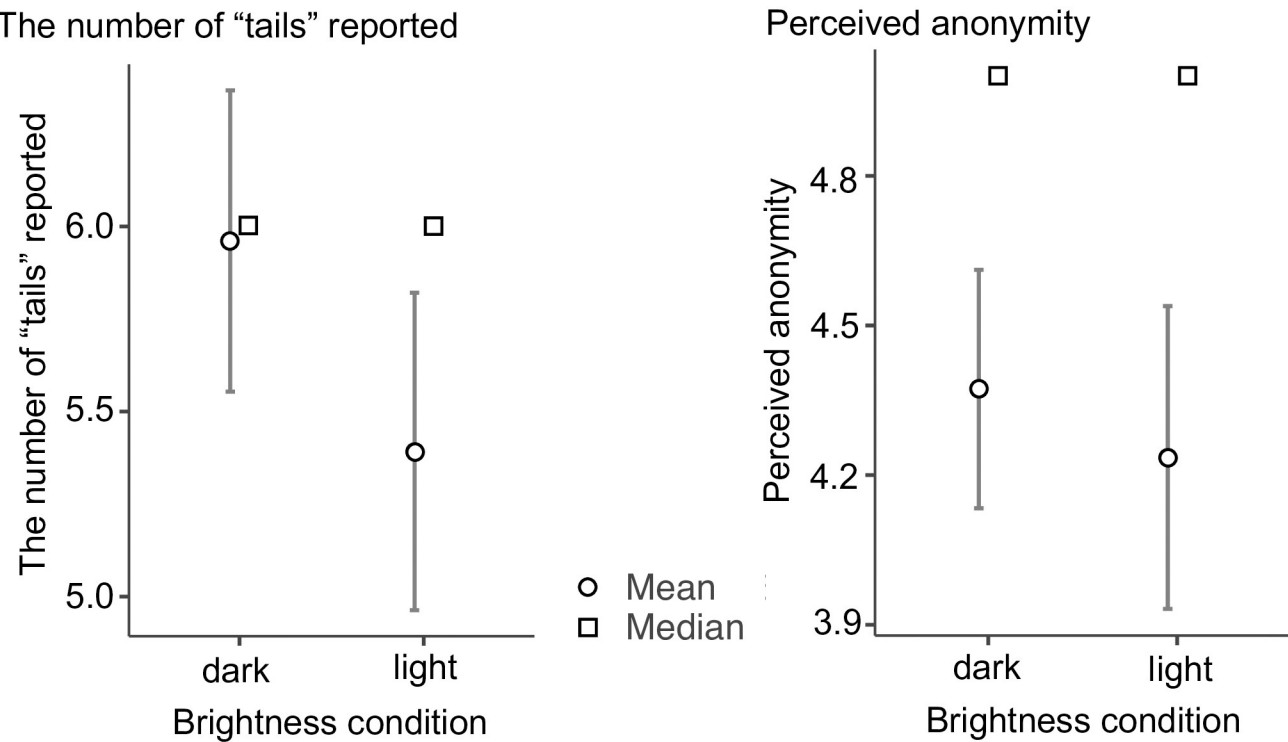

**Fig 2. Mean and median number of "tails" reported and perceived anonymity in the two conditions.** Note. Error bars denote 95% confidence intervals.

Moreover, the number of "tails" that participants reported in the two conditions was both significantly higher than chance level (dark condition: $t(50) = 4.63$, $p < .001$, Cohen's $d = 0.648$; light condition: $t(50) = 1.79$, $p = .040$, Cohen's $d = 0.251$).

## Discussion

By comparing the number of "tails" that participants reported in the two conditions, we found that the darkness of the room could promote participants' dishonesty. The result lends support to our hypothesis. This finding is also in line with the original experiment [9]. We successfully replicated the results of the original experiment, even though we implemented more precise control over the lighting conditions and utilized a different task. It is evident that the significant difference was found in means, but not in the medians. This discrepancy can be attributed to that some participants reported fairly higher scores ($> 7$) in the dark condition, whereas this was not the case in the light condition, with participants reporting scores that widely dispersed around 6. This indicates that dishonesty was more prevalent in the dark condition.

Besides, dishonesty was observed in both two conditions. It suggests that participants in the experiment have a tendency to be dishonest overall. This is similar to the results of Zhong et al. (2010) [9]. Moreover, taken as a whole, dishonest behavior is evident at a moderate level. This also demonstrates that the coin toss task and reward system we utilized provided participants with an appropriate motivation to lie, while also creating an environment in which participants felt free to be dishonest.

Moreover, the results of the manipulation check showed that the brightness used in this experiment had significant effects on participants' perceived brightness, which means the experimental manipulation were appropriate.

On the other hand, the more interesting finding pertained to perceived anonymity. Based on the results of TOST, the perceived anonymity of the participants in the two conditions was equal. That is to say, although we found that the darkness of the room did promote dishonesty, the cause of this phenomenon might not be a difference between participants' perceived anonymity as Zhong et al. (2010) inferred [9]. The difference between the results of this experiment and those of the original experiment could potentially be attributed to differences in experimental environment and design. For instance, in the original experiment, participants were not alone, while in this experiment, there was no presence of a second individual within the participant's field of view. Such difference undoubtedly impacts the participants' perceived anonymity. The influence of the presence of others on the actual anonymity of the experimental environment is also worth noting. Considering that despite the absence of others, participants in a laboratory experiment may still experience a sense of being observed, the issue will become more complex. In sum, under different actual levels of anonymity in experimental settings, the influence of brightness on perceived anonymity could vary and thus requires further empirical exploration. Based on the results of the present study, we considered that the conclusions drawn in the original study attributing the promoting effect of the darkness of the room on dishonesty to perceived anonymity may be presumptive, especially in circumstances where perceived anonymity was not thoroughly investigated.

Given the absence of the effects of perceived anonymity in this study, we would like to propose another plausible explanation. The results of our research can be interpreted within the framework of Conceptual Metaphor Theory (CMT), as introduced by Lakoff and Johnson (1980) [20]. CMT was defined as follows: a conceptual metaphor is understanding one domain of experience that is typically abstract in terms of another that is typically concrete [21]. Beyond conceptual understanding and emotional expression, psychological studies have demonstrated that human behaviors and moral concepts can also be influenced by metaphors. Frank and Gilovich (1988) reported black uniforms increased aggressiveness [22]. Sherman and Clore (2009) found that comparing to the situations that color and morality of the words do not match, the speed of color naming was faster when words in black concerned immorality and words in white concerned morality [23]. Banerjee et al. (2012) found that participants who recall and describe in detail an unethical deed from their past judged the room to be darker than the participants who recall an ethical deed [24]. This line of evidence proposes an automatic association of black (dim)-immoral and white (light)- moral metaphor in human cognition. Consequently, we hypothesize that the occurrence of dishonest behavior may vary when environmental stimuli are potent enough to evoke the black-white metaphor. In this respect, if the darkness in Zhong et al. (2010) or this experiment could possibly have evoked a "black" metaphor among the participants, this might have facilitated dishonest behavior.

A further alternative view of the relation between darkness and dishonesty can be offered by the latest findings from social psychology. It has been reported that the effect of brightness on pro-social actions may be moderated by societal factors, such as self-construal [25]. Moreover, the influence of brightness on pro-social behavior is believed to be mediated, in part, by satisfaction with illumination and perceived anonymity [26]. Intriguingly, the findings on perceived anonymity by Ru et al. (2022) differ from those of the present study. They observed that variations in brightness affected participants' perceived anonymity. It is postulated that this discrepancy may arise from the subtle differences in the design of the laboratory environment and the positioning of the experimenter. However, it is important to note that although pro-social actions and dishonest behavior are often considered in tandem within behavioral ethics, their subtle distinctions (e.g., dishonest behavior frequently entails the possibility of more stringent penalties) make it debatable to what extent findings on pro-social behavior can be reliably applied to dishonest actions. In summary, these suggest the presence of intricate

underlying mechanisms and multifaceted phenomena awaiting exploration in subsequent studies.

Lastly, there are other potential issues that should be acknowledged as constraints on generalization (COG, [27]). Importantly, these considerations should be kept in mind for any subsequent studies aiming to replicate the findings of this experiment, thereby enhancing the success of such replication efforts. Considering the observed differences in dishonest behavior between males and females [28–30], the unbalanced male-to-female ratio (29.41% vs. 68.63%) in this study could potentially have influenced the results of the experiment. Furthermore, the reward in this study was carefully determined, with consideration of the attractiveness to the participant group. Given that monetary gain was considered as an important factor prompting participants to be dishonest in this experiment and considering that rewards were shown to influence dishonest behavior [31, 32], the deliberation of an appropriate reward amount may become a particularly crucial aspect for future research. Additionally, the potential impact of other factors (e.g., the nationality of the participants, the visibility of the experimental place) should also be considered in future replication studies.

## Conclusions

We successfully replicated the original experiment that darkness was found to promote dishonest behavior, albeit with more precise control over the lighting conditions, and by using another task. However, our results of perceived anonymity did not align with the interpretation of the original study. Consequently, we posit that the conclusions drawn in the original experiment may have been somewhat presumptive and suggest further empirical validation. Future studies need to investigate potential explanations by metaphorical factors.

## Acknowledgments

We would like to thank Yanqing Tang, Linzi Liu for the assistance in preparing ethical review documents and data collection.

## Author Contributions

**Conceptualization:** Huanxu Liu, Yuki Yamada.

**Data curation:** Huanxu Liu.

**Formal analysis:** Huanxu Liu, Yuki Yamada.

**Funding acquisition:** Huanxu Liu, Yuki Yamada.

**Investigation:** Huanxu Liu, Yuning Zhou.

**Methodology:** Huanxu Liu, Yuki Yamada.

**Project administration:** Yuki Yamada.

**Resources:** Yuning Zhou, Yuki Yamada.

**Supervision:** Yuki Yamada.

**Validation:** Huanxu Liu, Yuki Yamada.

**Visualization:** Huanxu Liu.

**Writing – original draft:** Huanxu Liu, Yuki Yamada.

**Writing – review & editing:** Huanxu Liu, Yuning Zhou, Yuki Yamada.

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
