## [Decision Letter · Decision Letter 0]

20 Oct 2023

PONE-D-23-27810Darkness promotes dishonesty in a coin toss task: A pre-registered conceptual replication of Experiment 1 of Zhong, Bohns, and Gino (2010)PLOS ONE

Dear Dr. Liu,

Thank you for submitting your manuscript to PLOS ONE. After careful consideration, we feel that it has merit but does not fully meet PLOS ONE’s publication criteria as it currently stands. Therefore, we invite you to submit a revised version of the manuscript that addresses the points raised during the review process.

We look forward to receiving your revised manuscript.

Kind regards,

Guy Hochman, Ph.D.

Academic Editor

PLOS ONE

Journal Requirements:

2. Please expand the acronym “JST SPRING” (as indicated in your financial disclosure) so that it states the name of your funders in full.

"We would like to thank Yanqing Tang, Linzi Liu for the assistance in preparing ethical review documents and data collection. This study was financially supported by JST SPRING (https://www.jst.go.jp/jisedai/en/index.html), Grant Number JPMJSP2136 (HL), Grants-in-Aid for Scientific Research of the Japan Society for the Promotion of Science (https://www.jsps.go.jp/j-grantsinaid/index.html), Grant Numbers 22K18263, 21H03784, and 20H04581 (YY). The funders had no role in the study design, data collection and analysis, publication decision, or manuscript preparation."

"This study was financially supported by JST SPRING (https://www.jst.go.jp/jisedai/en/index.html), Grant Number JPMJSP2136 (HL), Grants-in-Aid for Scientific Research of the Japan Society for the Promotion of Science (https://www.jsps.go.jp/j-grantsinaid/index.html), Grant Numbers 22K18263, 21H03784, and 20H04581 (YY). The funders had no role in the study design, data collection and analysis, publication decision, or manuscript preparation."

6. We notice that your supplementary figures are uploaded with the file type 'Figure'. Please amend the file type to 'Supporting Information'. Please ensure that each Supporting Information file has a legend listed in the manuscript after the references list.

7. Please upload a copy of Figure 1 and 2, to which you refer in your text on page 6 and 9. If the figure is no longer to be included as part of the submission please remove all reference to it within the text.

8. Please include a separate caption for each figure in your manuscript.

Reviewers' comments:

Reviewer's Responses to Questions

**Comments to the Author**

1. Is the manuscript technically sound, and do the data support the conclusions?

Reviewer #1: Yes

Reviewer #2: Partly

2. Has the statistical analysis been performed appropriately and rigorously? 

Reviewer #1: Yes

Reviewer #2: Yes

3. Have the authors made all data underlying the findings in their manuscript fully available?

Reviewer #1: Yes

Reviewer #2: Yes

4. Is the manuscript presented in an intelligible fashion and written in standard English?

Reviewer #1: Yes

Reviewer #2: Yes

5. Review Comments to the Author

Reviewer #1: This seems to me a good replication study with pre-registration, power analysis, and quantifiable manipulation of light/darkness condition. The use of the dice task also extends generalizability. The absence of the anonymity difference does not seem too surprising to me -- I would expect perceived anonymity to be likely sensitive to experimental setup and the presence and positioning of other individuals. It would be helpful for the authors to report means and SDs in addition to statistical tests so we can see whether there might be floor or ceiling effects. Overall I think this is a worthy effort that should be published.

Reviewer #2: In the first line of the introduction, it is asserting that dishonesty is pervasive. Is it? Most bad and socially disapproved behavior is not normally distributed (e.g., Kim B. Serota et al.’s numerous studies of the few prolific liars). I also noted that in the current data, there did not appear to be pervasive cheating.

It would be helpful to report both means and standard deviations in text especially for the number of tails.

Why are the means different but not the medians in the tails. Do outliers explain the results?

The difference in findings between means and medians and the overall level/extent of cheating should be mentioned in the discussion section.

I really liked the simplicity of the experiment.

6. PLOS authors have the option to publish the peer review history of their article (what does this mean?). If published, this will include your full peer review and any attached files.

Reviewer #1: No

Reviewer #2: No

---

## [Author Response · Author response to Decision Letter 0]

22 Oct 2023

1) The Please ensure that your manuscript meets PLOS ONE's style requirements, including those for file naming. The PLOS ONE style templates can be found at　https://journals.plos.org/plosone/s/file?id=wjVg/PLOSOne_formatting_sample_main_body.pdf and　https://journals.plos.org/plosone/s/file?id=ba62/PLOSOne_formatting_sample_title_authors_affiliations.pdf.

Reply: We have carefully checked the template you provided and made necessary adjustments to the manuscript in accordance with it. We have also labeled the files in this submission as you asked.

2) Please expand the acronym “JST SPRING” (as indicated in your financial disclosure) so that it states the name of your funders in full. This information should be included in your cover letter; we will change the online submission form on your behalf.

Reply: The full designation for 'JST SPRING' stands for the 'Japan Science and Technology Agency, Support for Pioneering Research Initiated by the Next Generation'. Furthermore, the revised Financial Disclosure Statement is as follows:

“This work was supported by Japan Science and Technology Agency, Support for Pioneering Research Initiated by the Next Generation (https://www.jst.go.jp/jisedai/en/index.html), Grant Number JPMJSP2136 (HL), Grants-in-Aid for Scientific Research of the Japan Society for the Promotion of Science (https://www.jsps.go.jp/j-grantsinaid/index.html), Grant Numbers 22K18263, 21H03784, and 20H04581(YY). The funders had no role in the study design, data collection and analysis, publication decision, or manuscript preparation.”

3) Thank you for stating the following in the Acknowledgments Section of your manuscript: "We would like to thank Yanqing Tang, Linzi Liu for the assistance in preparing ethical review documents and data collection. This study was financially supported by JST SPRING (https://www.jst.go.jp/jisedai/en/index.html), Grant Number JPMJSP2136 (HL), Grants-in-Aid for Scientific Research of the Japan Society for the Promotion of Science (https://www.jsps.go.jp/j-grantsinaid/index.html), Grant Numbers 22K18263, 21H03784, and 20H04581 (YY). The funders had no role in the study design, data collection and analysis, publication decision, or manuscript preparation."

Please remove any funding-related text from the manuscript and let us know how you would like to update your Funding Statement. Currently, your Funding Statement reads as follows: "This study was financially supported by JST SPRING (https://www.jst.go.jp/jisedai/en/index.html), Grant Number JPMJSP2136 (HL), Grants-in-Aid for Scientific Research of the Japan Society for the Promotion of Science (https://www.jsps.go.jp/j-grantsinaid/index.html), Grant Numbers 22K18263, 21H03784, and 20H04581 (YY). The funders had no role in the study design, data collection and analysis, publication decision, or manuscript preparation."

Reply: We deeply regret the oversight on my part. We have amended the acknowledgements section of the manuscript as directed, deleted all text related to funding. The amended statements were attached to Reply 2) above. Thank you.

4) We note that you have stated that you will provide repository information for your data at acceptance. Should your manuscript be accepted for publication, we will hold it until you provide the relevant accession numbers or DOIs necessary to access your data. If you wish to make changes to your Data Availability statement, please describe these changes in your cover letter and we will update your Data Availability statement to reflect the information you provide.

Reply: Thank you for bringing this to my attention. We have revised the content of the statement and incorporated a link that can access the data. The updated statement reads as follows: “The datasets analyzed in this study are available from the OSF repository (osf.io/bvqsn).”

5) Your ethics statement should only appear in the Methods section of your manuscript. If your ethics statement is written in any section besides the Methods, please move it to the Methods section and delete it from any other section. Please ensure that your ethics statement is included in your manuscript, as the ethics statement entered into the online submission form will not be published alongside your manuscript.

Reply: Thank you for drawing my attention to this matter. I have incorporated the ethics statement into the methods section and duly removed it from other section.

6) We notice that your supplementary figures are uploaded with the file type 'Figure'. Please amend the file type to 'Supporting Information'. Please ensure that each Supporting Information file has a legend listed in the manuscript after the references list.

Reply: My sincerest apologies for any confusion. It appears there may have been a misunderstanding on my part regarding your publication's guidelines. I would like to clarify that these are “figures” not “supplementary figures”. Consequently, I have made appropriate revisions to the manuscript to reflect this. They will be uploaded with the file type “Figure” in this submission. Should you consider it would be more fitting for them to be submitted as supplementary figures or have any other concerns, please do not hesitate to inform me. I am more than willing to make further adjustments as necessary.

7) Please upload a copy of Figure 1 and 2, to which you refer in your text on page 6 and 9. If the figure is no longer to be included as part of the submission please remove all reference to it within the text.

Reply: As previously mentioned, I mistakenly submitted Figure 1 and 2 as supplementary figures. I deeply regret any confusion caused. I have addressed this in the revised manuscript. 

8) Please include a separate caption for each figure in your manuscript.

Reply: I apologize again for the oversight that submitted the figures as supplementary figures. I have ensured the revised manuscript meets your journal's requirements concerning figure presentation. Specifically, the misplaced captions have been removed from the end of the manuscript and have been appropriately positioned where Figures 1 and 2 are first cited.

Reply: We have diligently reviewed and adjusted the reference list to meet the standards of your journal. To the best of our knowledge, there are no retracted papers.

# Review Comments to the Author

Reviewer #1: This seems to me a good replication study with pre-registration, power analysis, and quantifiable manipulation of light/darkness condition. The use of the dice task also extends generalizability. The absence of the anonymity difference does not seem too surprising to me -- I would expect perceived anonymity to be likely sensitive to experimental setup and the presence and positioning of other individuals. It would be helpful for the authors to report means and SDs in addition to statistical tests so we can see whether there might be floor or ceiling effects. Overall I think this is a worthy effort that should be published.

Reply: Thank you for your positive feedback. We are gratified to note that you recognize the importance and implications of our study. Additionally, we are in agreement with your insights on anonymity. Per your recommendation, we have provided further details to the readers in the results section, such as means and SDs. 

“The mean score reported in dark condition was 5.96 (SD = 1.48; Mdn = 6); the mean score reported in light condition was 5.39 (SD = 1.56; Mdn = 6).” 

“On the other hands, the mean perceived anonymity in dark condition was 4.37 (SD = 0.87), while that in light condition was 4.24 (SD = 1.11).”

Reviewer #2: In the first line of the introduction, it is asserting that dishonesty is pervasive. Is it? Most bad and socially disapproved behavior is not normally distributed (e.g., Kim B. Serota et al.’s numerous studies of the few prolific liars). I also noted that in the current data, there did not appear to be pervasive cheating.

Reply: As you rightly pointed out, our phrasing could be misunderstandings. We initially aimed to convey that dishonest behavior often occurs in daily life. We have revised the opening sentence of the introduction to ensure clarity and avoid potential misunderstandings, “The potential adverse consequences of dishonesty in daily life have catalyzed extensive research in psychology”. We deeply appreciate your insightful comments, and we're grateful for the information you provided.

Reviewer #2: It would be helpful to report both means and standard deviations in text especially for the number of tails.

Reply: Thank you for your feedback, which guided us to enhance our presentation of results. In response, we have provided the means and standard deviations of the number of tails in the revised manuscript.

“The mean score reported in dark condition was 5.96 (SD = 1.48; Mdn = 6); the mean score reported in light condition was 5.39 (SD = 1.56; Mdn = 6).”

Reviewer #2: Why are the means different but not the medians in the tails. Do outliers explain the results?

Reply: That is an interesting question. We consider that the reason lies in the outliers as you have mentioned, particularly those participants reporting high tails. The distributions of the individual data show that there were more participants in the dark condition who reported high tails. Precisely, that was the central conclusion of the experiment. Conversely, data from the light condition predominantly clustered around the value of 6, which makes a lower Mean. We added this explanation in the text.

“It is evident that the significant difference was found in means, but not in the medians. This discrepancy can be attributed to that some participants reported fairly higher scores (> 7) in the dark condition, whereas this was not the case in the light condition, with participants reporting scores that widely dispersed around 6. This indicates that dishonesty was more prevalent in the dark condition.”

Reviewer #2: The difference in findings between means and medians and the overall level/extent of cheating should be mentioned in the discussion section.

Reply: We greatly appreciate your insightful comment. In response, we have revised the manuscript to incorporate the content aligned with your suggestions.

“It is evident that the significant difference was found in means, but not in the medians. This discrepancy can be attributed to that some participants reported fairly higher scores (> 7) in the dark condition, whereas this was not the case in the light condition, with participants reporting scores that widely dispersed around 6. This indicates that dishonesty was more prevalent in the dark condition.”

“Moreover, taken as a whole, dishonest behavior is evident at a moderate level.”

Reviewer #2: I really liked the simplicity of the experiment.

Reply: We deeply appreciate your positive comment; it's truly our privilege.

---

## [Editor Report · Decision Letter 1]

30 Oct 2023

PONE-D-23-27810R1Darkness promotes dishonesty in a coin toss task: A pre-registered conceptual replication of Experiment 1 of Zhong, Bohns, and Gino (2010)PLOS ONE

Dear Dr. Liu,

Thank you very much for revising your manuscript based on the comments raised by the reviewers. While you addressed all these concerns, I still have some minor issues that should be addressed before I can formally accept your paper. Thus, I conditionally accept your manuscript, pending some minor issues that should increase the impact of your paper. Please try to address this issue the best you can.

First, on page 3, you refer to physiology research in dishonesty but only cite one source (that deals with hormones). Please add 1-2 papers that used physiological measures to study cheating and immoral behavior (whichever you like).

Second, when citing Zhang et al on page 4, l. 74-75, please add the page number to the end of the citation.

Third, referring to the second reason you state for replicating Zhang et al., please explain why it is important to examine perceived anonymity (how it adds to what is already known in the literature). Moreover, please try to emphasize your unique contribution to the literature on cheating behavior (as it is not clear and explicit enough).

Finally, please refer to up-to-date research when interpreting your results and develop the argument about the CMT. You might benefit from using the following references (but don’t feel obligated to use them). But please try your best to better explain why you find the difference between the light and dark conditions and how it relates to recent literature in behavioral ethics.

Esteky, S., Wooten, D. B., & Bos, M. W. (2020). Illuminating illumination: Understanding the influence of ambient lighting on prosocial behaviors. Journal of Environmental Psychology, 68, 101405.

Ru, T., Ma, Y., Zhong, L., Chen, Q., Ma, Y., & Zhou, G. (2022). Effects of Ambient Illuminance on Explicit and Implicit Altruism: The Mediation Roles of Perceived Anonymity and Satisfaction with Light. International Journal of Environmental Research and Public Health, 19(22), 15092.

We look forward to receiving your revised manuscript.

Kind regards,

Guy Hochman, Ph.D.

Academic Editor

PLOS ONE
---

## [Author Response · Author response to Decision Letter 1]

31 Oct 2023

Dr. Guy Hochman

Academic Editor

PLOS ONE

31, October 2023

Dear Dr. Hochman,

Thank you very much for handling our manuscript titled, “Darkness promotes dishonesty in a coin toss task: A pre-registered conceptual replication of Experiment 1 of Zhong, Bohns, and Gino (2010)” (PONE-D-23-27810R1). We have revised the manuscript according to your comments to improve the quality and to increase the impact of our paper.

We are very grateful for your efforts and comments on this manuscript. If there is still any problem, please feel free and let us know.

Below, we have outlined our point-by-point responses.

# Editor Comments

1) First, on page 3, you refer to physiology research in dishonesty but only cite one source (that deals with hormones). Please add 1-2 papers that used physiological measures to study cheating and immoral behavior (whichever you like).

Reply: We have incorporated the following references (about hormones and facial thermal variations) and subsequently refined the manuscript accordingly: “Physiological psychology has also contributed to the understanding of this issue, with the studies about the relation between physiological indicators (such as hormones and facial thermal variations) and dishonesty behavior [6, 7, 8]” 

“7. Wibral M, Dohmen T, Klingmüller D, Weber B, Falk A. Testosterone Administration Reduces Lying in Men. PLoS ONE. 2012;7(10): e46774. https://doi.org/10.1371/journal.pone.0046774”

“8. Rajoub BA, Zwiggelaar R. Thermal facial analysis for deception detection. IEEE Trans Inf Forensics Secur. 2014;9(6): 1015-1023.　https://doi.org/10.1109/TIFS.2014.2317309.”

2) Second, when citing Zhang et al on page 4, l. 74-75, please add the page number to the end of the citation.

Reply: Thank you for noting the problem. The specific page numbers have now been added. 

“They said that “although the room in the experimental condition was darker than the one in the control condition, participants had no trouble seeing and identifying each other” [9, p. 314].”

3) Third, referring to the second reason you state for replicating Zhang et al., please explain why it is important to examine perceived anonymity (how it adds to what is already known in the literature). Moreover, please try to emphasize your unique contribution to the literature on cheating behavior (as it is not clear and explicit enough).

Reply: Concerning the rationale for examining perceived anonymity, we have integrated the following sentences into the original manuscript. “Zhong et al. (2010) did not examine any measures related to perceived anonymity, thus it is unclear whether the mechanism in which room darkness leads to cheating really involves perceived anonymity. That is, the effect of anonymity has insufficient plausible grounds to support the conclusions of the original study. Further empirical evidence is required to substantiate this claim.”

Concurrently, we have further underscored our strengths.

 “In summation, our refinements provide several key advantages: first, they can control the variability of actual anonymity. Second, they can provide evidence for the first time on perceived anonymity in the observed phenomenon. Third, they facilitate a discussion regarding the potential generalizability of the phenomenon, especially under tasks more suitable for detecting dishonest behavior.”

4) Finally, please refer to up-to-date research when interpreting your results and develop the argument about the CMT. You might benefit from using the following references (but don’t feel obligated to use them). But please try your best to better explain why you find the difference between the light and dark conditions and how it relates to recent literature in behavioral ethics.

Reply: Thank you very much for the information! Addressing the issues you point out, we have enriched our discussion section by citing the recent research you provided. “A further alternative view of the relation between darkness and dishonesty can be offered by the latest findings from social psychology. It has been reported that the effect of brightness on pro-social actions may be moderated by societal factors, such as self-construal [25]. Moreover, the influence of brightness on pro-social behavior is believed to be mediated, in part, by satisfaction with illumination and perceived anonymity [26]. Intriguingly, the findings on perceived anonymity by Ru et al. (2022) differ from those of the present study. They observed that variations in brightness affected participants’ perceived anonymity. It is postulated that this discrepancy may arise from the subtle differences in the design of the laboratory environment and the positioning of the experimenter. However, it is important to note that although pro-social actions and dishonest behavior are often considered in tandem within behavioral ethics, their subtle distinctions (e.g., dishonest behavior frequently entails the possibility of more stringent penalties) make it debatable to what extent findings on pro-social behavior can be reliably applied to dishonest actions. In summary, these suggest the presence of intricate underlying mechanisms and multifaceted phenomena awaiting exploration in subsequent studies.”

# Journal Requirements

Reply: We have diligently reviewed and adjusted the reference list to meet the standards of PLOS ONE. To the best of our knowledge, there are no retracted papers.

The manuscript has been revised according to these replies. We are very grateful for your efforts and comments on this manuscript. If there is still any problem, please feel free and let us know.

Sincerely,

Huanxu Liu

Graduate School of Human-Environment Studies, Kyushu University,

744 Motooka, Nishi-ku, Fukuoka 819-0395 Japan

E-mail: ryuukansyo@gmail.com

TEL & FAX: +81-92-802-5837

Yuki Yamada, Ph.D.

Faculty of Arts and Science, Kyushu University,

744 Motooka, Nishi-ku, Fukuoka 819-0395, Japan

E-mail: yamadayuk@gmail.com

TEL & FAX: +81-92-802-5837

---

## [Editor Report · Decision Letter 2]

2 Nov 2023

Darkness promotes dishonesty in a coin toss task: A pre-registered conceptual replication of Experiment 1 of Zhong, Bohns, and Gino (2010)

PONE-D-23-27810R2

Dear Dr. Liu,

We’re pleased to inform you that your manuscript has been judged scientifically suitable for publication and will be formally accepted for publication once it meets all outstanding technical requirements.

Kind regards,

Guy Hochman, Ph.D.

Academic Editor

PLOS ONE
---

## [Editor Report · Acceptance letter]

16 Nov 2023

PONE-D-23-27810R2 

Darkness promotes dishonesty in a coin toss task:
A pre-registered conceptual replication of Experiment 1 of Zhong, Bohns, and Gino (2010) 

Dear Dr. Liu:

I'm pleased to inform you that your manuscript has been deemed suitable for publication in PLOS ONE. Congratulations! Your manuscript is now with our production department. 

Kind regards, 

on behalf of

Professor Guy Hochman 

Academic Editor

PLOS ONE